



# Comparisons of Planetary Boundary Layer Height from Ceilometer with ARM Radiosonde Data

Damao Zhang, Jennifer Comstock, Victor Morris

Pacific Northwest National Laboratory, Richland, WA, USA

*Correspondence to*: Damao Zhang (damao.zhang@pnnl.gov)

**Abstract.** Ceilometer measurements of aerosol backscatter profiles have been widely used to provide continuous PBLHT estimations. To investigate the robustness of ceilometer-estimated PBLHT under different atmospheric conditions, we compared ceilometer- and radiosonde-estimated PBLHTs using long term U.S. Department of Energy (DOE) Atmospheric

Radiation Measurement (ARM) ceilometer and balloon-borne sounding data at three ARM fixed-location atmospheric observatories and from three ARM mobile observatories deployed around the world for various field campaigns, which cover from Tropics to Polar regions and over both ocean and land surfaces. Statistical comparisons of ceilometer-estimated PBLHTs from the Vaisala CL31 ceilometer data with radiosonde-estimated PBLHTs from the ARM PBLHT-SONDE Value-added Product (VAP) are performed under different atmospheric conditions including stable and unstable atmospheric boundary

layer, low-level cloud-free, and cloudy conditions at these ARM observatories. Under unstable atmospheric boundary layer conditions, good comparisons are found between ceilometer- and radiosonde-estimated PBLHTs at ARM low- and mid-latitude land observatories. However, it is still challenging to obtain reliable PBLHT estimations over ocean surfaces even using radiosonde data. Under stable atmospheric boundary layer conditions, ceilometer- and radiosonde-estimated PBLHTs have weak correlations. Among different PBLHT estimations utilizing the Heffter, the Liu-Liang, and the bulk Richardson

number methods in the ARM PBLHT-SONDE VAP, ceilometer-estimated PBLHTs have better comparisons with the Liu-Liang method under unstable and with the bulk Richardson number method under stable atmospheric boundary layer conditions.

## 1 Introduction

The planetary boundary layer is the lowest part of the troposphere that directly interacts with the earth's surface. The effects

of surface friction, heating, and cooling cause significant exchanges of heat, mass, moisture, and momentum between the planetary boundary layer and the earth's surface through turbulent motions (Stull 1988). Therefore, the planetary boundary layer structure responds quickly to surface forcing and may have large temporal and spatial variations, especially over land (Seidel et al., 2010; von Engeln and Teixeira 2013). Planetary boundary layer height (PBLHT) is a key parameter that characterizes the structure of the lowest few kilometers of the atmosphere and has a great impact on air quality, global climate,

land-atmosphere interactions, and a wide range of atmospheric processes such as cloud formation and evolution, aerosol mixing and transport, and aerosol-cloud interactions (Seinfeld et al., 2006; Konor et al., 2009; Lemone et al., 2018).



Following Stull (1988) and Liu and Liang (2010), the boundary layer structure can be classified into three major regimes depending on the atmospheric thermodynamic environment: convective boundary layer (CBL), stable boundary layer (SBL), and residual layer (RL). Under the CBL condition, which generally occurs during the daytime, the strong turbulence and convection causes intense mixing within the boundary layer. The top of the boundary layer is often characterized by an inversion layer of potential temperature and a pronounced decrease of moisture and pollutant concentration. For deep CBLs such as in the Tropics, however, it might be difficult to determine the top of the boundary layer using the potential temperature inversion (Kepert et al., 2016). Under the SBL condition, turbulence tends to be suppressed by the statically stable air above it and occurs only sporadically. The PBLHT is defined as the top of the stable layer or the height where turbulence is negligible compared to its surface value (Stull 1988). The RL is usually formed during the evening or morning transition time. A RL that is associated with near-neutral conditions in the surface layer is neutrally stratified and keeps similar state variables and pollutant profiles as the recently decayed CBL and is referred to as the neutral residual layer (NRL) hereafter. It should be noted that atmospheric boundary layer stability ranges from very stable to strongly unstable. Classification of atmospheric boundary layer stability into these three major regimes is simplified and may not be appropriate for transient atmospheric conditions (Mahrt 1999).

The PBLHT in atmospheric models is usually calculated by using either diagnostic equations that take surface fluxes and the initial temperature profile as inputs or by using the Richardson number profile to find the first level where the Richardson number exceeds a critical value (Seibert et al., 2000). On the other hand, from observational studies, the PBLHT has been widely determined using radiosonde data that provide profiles of atmospheric temperature, pressure, and moisture (Seibert et al., 2000; Liu and Liang 2010, Seidel et al., 2010). Methods using the elevated temperature inversion, or the maximum vertical gradient of potential temperature, or minimum vertical gradient of moisture, or surface-based inversion to determine PBLHT under different regimes have been developed (Stull 1988; Bradley et al., 1993; Seidel et al., 2010). For example, both Heffter (1980) and Liu and Liang (2010) use potential temperature gradient as a key parameter to determine the PBLHT for CBL and NRL regimes. However, radiosonde data have poor temporal resolutions and are subject to sampling errors. Most radiosonde stations launch a sounding system twice daily and thus cannot capture the diurnal evolution of the PBLHT (Seidel et al., 2010). Observing atmospheric boundary layer transitions with high temporal-spatial resolutions is required to investigate atmospheric thermodynamic processes (Fritz et al., 2021).

Remote sensing systems such as sodars, radio-acoustic sounding systems, wind profiling radars, and lidars provide high-temporal continuous observations that can be used to estimate PBLHT (Seibert et al., 2010). Especially, aerosol lidar systems measuring vertical aerosol backscatter profiles with high temporal and vertical resolutions have also been widely used to derive PBLHT in recent years (Steyn et al., 1999; Brooks 2003; Sawyer and Li 2013; Dang et al., 2019; Su et al., 2020). Space-borne lidar such as the Cloud-Aerosol Lidar with Orthogonal Polarization (CALIOP) onboard the Cloud-Aerosol Lidar and Infrared



Pathfinder Satellite Observation (CALIPSO) satellite can even provide a global PBLHT climatology, although it is unable to capture the diurnal cycle (Luo et al., 2016). Aerosol lidars use aerosol as tracers and a gradient of aerosol backscatter signals is generally used to derive the PBLHT. Numerous methods that use a prescribed lidar backscatter signal threshold (Frioud et al., 2003), first and second derivative of lidar signals (Sicard et al., 2006; Luo et al., 2014), lidar signal wavelet transformation

(Brooks 2003; Davis et al., 2005), and curve-fitting to idealized lidar profiles (Steyn et al., 1999) have been proposed to estimate the PBLHT.

A laser ceilometer is a type of atmospheric lidar that measures backscattered laser signals from atmospheric particles such as aerosols and cloud droplets. Particularly, laser ceilometers are low-cost and reliable systems that provide fully automated all-

weather measurements. Laser ceilometers have been deployed over many locations around the world and their measurements have been widely used for cloud base detections and atmospheric aerosol and cloud structure analyses (Martucci et al., 2010; Kotthaus and Grimmond 2018). To take the advantage of those continuous long-term ceilometer measurements, several PBLHT retrieval techniques using ceilometer aerosol backscatter data have been adopted to study the characteristics and evolutions of the boundary layer at various locations and to monitor the temporal and spatial variations of PBLHT (Münkel et

al., 2007; Caicedo et al. 2017). Evaluations from previous studies show good agreement between PBLHT derived from ceilometer and radiosonde data for limited cases (Haeffelin et al., 2012; Haman et al., 2012). However, those evaluations are based on limited data from a single location or a short-term campaign, the robustness of the estimated PBLHT from laser ceilometer measurements has not yet been validated under various atmospheric conditions and over multiple locations with different surface properties.


In this study, we use long-term US Department of Energy (DOE) Atmospheric Radiation Measurement (ARM) ground-based remote sensing measurements and balloon-borne sounding data to compare and evaluate PBLHT estimated from ceilometer backscatter with ARM sounding data at three ARM fixed-location atmospheric observatories and three ARM mobile facilities (AMFs) deployed around the world for various field campaigns. Long-term data at these climatologically significant locations

allow us to statistically investigate how surface properties impact PBLHT estimation methods, how well PBLHT estimation methods perform under different atmospheric boundary layer regimes, and PBLHT diurnal and seasonal variations. The paper is organized as follows: Section 2 presents a brief description of ARM ground-based remote sensing measurements and methodologies used to derive PBLHT from sounding data and ceilometer measurements. Section 3 shows statistical comparisons of ceilometer- and radiosonde-estimated PBLHTs under different atmospheric conditions including stable and

unstable atmospheric boundary layers, low-level cloud-free, and cloudy conditions at various ARM observatories. PBLHT diurnal evolution and its seasonal variations are also presented. Summary and conclusions are given in section 4.



## 2 Datasets and Methodology

The DOE ARM user facility provides continuous field measurements of atmospheric conditions by deploying remote sensing and *in situ* atmospheric observatories at climatically significant locations. ARM operates three fixed-location atmospheric observatories at U.S. Southern Great Plains (SGP), North Slope of Alaska (NSA), and Eastern North Atlantic (ENA) located in the Azores. These fixed-location observatories have been acquiring long-term measurements of cloud, aerosol, precipitation, and atmospheric dynamic and thermodynamic data for over 25-years at some locations. In this study, we also use data from the former ARM Tropical Western Pacific (TWP) observatory located at Darwin, Australia. In addition, ARM operates three mobile facilities (AMFs) which can be requested by scientists through a proposal process for various field campaigns that deploy ARM instruments anywhere in the world for roughly a year. We use observations from five AMF Field Campaigns including the Observations and Modeling of the Green Ocean Amazon – GOAMAZON (MAO), the Layered Atlantic Smoke Interactions with Clouds – LASIC (ASI), the Cloud, Aerosol, and Complex Terrain Interactions – CACTI (COR), the ARM West Antarctic Radiation Experiment – AWARE (AWR), and the AMF3 deployment at the Oliktok Point (OLI). The three-letter ARM identifier in parenthesis is defined by a geographic reference or the International Air Transport Association (IATA) three-letter airport code to indicate approximate location. The geographical locations of ARM fixed-location observatories and AMF field campaign deployments used in this study are shown in Figure 1. Table 1 lists the site elevations above sea level (ASL), surface characteristics, time periods, and the number of radiosondes for ARM observatories and AMF deployments.

ARM places various state-of-the-art instrument platforms at each observatory including radiometers, radars, lidars (including ceilometers), total sky imagers, surface meteorological instrumentation, aerosol observing systems, and radiosondes. Details on these instruments and their measurements are presented in Mather and Voyles (2013) and can also be found from each ARM instrument handbook (https://www.arm.gov/capabilities/instruments). Furthermore, ARM produces higher-order data products named Value-Added Products (VAP) using existing ARM datastreams as inputs. VAPs use quality-controlled data to derive higher-order atmospheric quantities that can be more directly used for atmospheric research and by global climate models. A full list of ARM VAPs can be found at the ARM VAP website (https://www.arm.gov/capabilities/vaps). All the data obtained at ARM fixed-location observatories and AMF field campaigns and derived VAPs are available at the ARM Data Discovery website (https://adc.arm.gov/discovery). In this study, we mainly focus on analyses of ARM radiosonde data, ceilometer measurements, and corresponding VAPs.

Each ARM observatory generally launches four times a day (except twice daily at NSA and OLI) the balloon-borne sounding system (SONDE) at roughly 5:30, 11:30, 17:30, and 23:30 Universal Time Coordinate (UTC) (5:30 and 17:30 UTC at NSA and 17:30 and 23:30 UTC at OLI). SONDE provides measurements of vertical profiles of atmospheric thermodynamic state such as atmospheric pressure, temperature, moisture, and the wind speed and direction with a 1-second temporal resolution, corresponding to vertical height resolutions of several meters to more than 10 meters depending on the atmospheric dynamic



environment. The accuracies of radiosonde measured temperature and wind speed are 0.2 °C and 0.2 m/2, respectively. The ARM PBLHT-SONDE VAP implements three commonly used methods developed by Heffter (1980, referred to as the Heffter method hereafter), Liu and Liang (2010, referred to as the Liu-Liang method hereafter), and Seibert et al. (2000, referred to as the bulk Richardson number method hereafter) to estimate PBLHT from radiosonde data (Sivaraman et al., 2013). The Heffter method determines the PBLHT from a potential temperature gradient profile. It first identifies each potential temperature

inversion layer, which is defined as two or more continuous heights where the potential temperature lapse rate is greater than 0.005 K/m. PBLHT is then determined as the base height of the lowest inversion layer in which the potential temperature difference between the base and top of the inversion layer is greater than 2 K. If the algorithm does not identify an inversion layer below 4 km above the ground level (AGL) that meets the criteria, the Heffter method identifies the base height of the inversion layer that has the largest potential temperature gradient within 4 km AGL as the PBLHT and flags the derived

PBLHT as indeterminate.

The Liu-Liang method uses different algorithms for different boundary layer structure regimes to determine PBLHT. The first step is to identify the boundary layer regime using the potential temperature ($\theta$) difference between the fifth and second level of sounding data ($\theta_5$ - $\theta_2$). The Liu-Liang method classifies the boundary layer regime as CBL, SBL, or NRL by comparing $\theta_5$

- $\theta_2$ with a stability threshold $\delta_s$. For the CBL and NRL regimes, the PBLHT is determined following Stull (1988) as the height at "which an air parcel rising adiabatically from the surface becomes neutrally buoyant". Practically, the Liu-Liang method searches upwardly for the PBLHT as the level $k$ at which $\theta_k$ - $\theta_2$ > $\delta_u$, where $\delta_u$ is another stability threshold, and the potential temperature gradient ($\dot{\theta}_k$) is larger than a gradient threshold $\dot{\theta}_r$. The threshold values of $\delta_s$, $\delta_u$, and $\dot{\theta}_r$ are dependent on the surface type and are empirically determined in Liu and Liang (2010). For the SBL regime, however, the determination of the

PBLHT is much more challenging as the SBL turbulence can result from either buoyancy forcing generated by the stable layer above the surface or wind shear that is usually associated with low-level jet (LLJ). The Liu-Liang method determines the PBLHT for the SBL regime as the top of the stable layer above the surface or the height of the LLJ nose, whichever is lower.

The bulk Richardson number $Ri$ represents the ratio of thermally produced turbulence to that generated by vertical wind shear.

Since wind shear produced turbulence is greatly reduced above the top of atmospheric boundary layer, $Ri$ increases dramatically at the top of SBL. The PBLHT is determined as the level at which $Ri$ is first greater than a critical value $Ri_c$. According to Sørensen et al. (1998), $Ri$ at a given altitude can be calculated from sounding data with the following equation:

$$Ri = \left(\frac{gz}{\theta_{v0}}\right)\left(\frac{\theta_{vz} - \theta_{v0}}{u_z^2 + v_z^2}\right) \tag{1}$$

where $g$ is the gravitational acceleration, $z$ is the height in AGL, $\theta_{vz}$ and $\theta_{v0}$ are the virtual potential temperatures at the surface

and height $z$, and $u_z$ and $v_z$ are the wind speed components at height z. The magnitude of $Ri_c$ employed in previous studies ranged from 0.25 to 0.5 (Mahrt 1981; Holtslag et al., 1990). Seibert et al. (2000) suggests an optimal $Ri_c$ value of 0.25 when



applied to radiosonde data. The ARM PBLHT-SONDE VAP provides estimated PBLHTs using two $Ri_c$ values of 0.25 and 0.5 (referred to as PBLHT Richardson_p25 and PBLHT Richardson_p5, correspondently).

Each PBLHT estimation is given a quality control (QC) flag to indicate possible issues that are related to input data files or unreasonable estimations (e.g., estimated PBLHT > 4 km AGL). Only PBLHT estimations with clear QC flags are selected in this study. It should be noted that different algorithms used in the PBLHT-SONDE VAP could produce dramatically different PBLHT estimations, especially for the SBL regime. The challenge is that there is no ground truth measurement to determine which method performs better than others. The performance of each method depends on the surface type and boundary layer

conditions. For example, previous studies suggested PBLHT estimated with the Liu-Liang method generally agrees better with lidar estimations than the Heffter and bulk Richardson number methods (Sawyer and Li 2013; Su et al., 2020). However, Lewis (2016) argued that the Liu-Liang and bulk Richardson number methods did not produce realistic PBLHT estimations while the Heffter method produces reasonable PBLHT values based on careful inspection of temperature and humidity profiles during the Marine ARM GPCI Investigation of Clouds (MAGIC) field campaign. Although the Heffter and Liu-Liang methods

generally provide more reliable PBLHT estimations for the CBL and NRL regimes, the bulk Richardson number method provides better PBLHT estimations for the SBL regime (Seibert et al., 2000).

ARM ceilometers use the Vaisala CL31 model, which has a maximum vertical range of 7.7 km (Münkel and Räsänen, 2004). The Vaisala CL31 ceilometer detects up to three cloud layers simultaneously and measures vertical visibility. In addition, the

Vaisala CL31 ceilometer also provides total attenuated backscatter coefficient profiles at the wavelength of 910 nm with a vertical resolution of 10 m and temporal resolution of 2 s, which have been used widely to derive continuous estimations of PBLHT (Münkel et al., 2007). To estimate PBLHT, the Vaisala CL31 ceilometer employs the gradient method that searches for local gradient minima of the range and overlap corrected total backscatter coefficient profile. To get more reliable aerosol signals, ceilometer data are first averaged to a temporal resolution of 16 s. To search for local gradient minima of the total

backscatter coefficient profile, ceilometer data are further applied with 30 minutes temporal and 360 m vertical sliding average. The presence of cloud layers and precipitation might impact PBLHT estimations, therefore, the enhanced gradient method applies a cloud and precipitation filter in the averaging process and suppresses false layer identification, which allows for robust estimations of PBLHT under all weather conditions (Münkel and Roininen, 2010). The Vaisala CL31 incorporates the enhanced gradient method into a built-in software called 'BL-VIEW' that provides real-time monitoring of boundary layer

structures and identifies up to three boundary layer height candidates. The BL-VIEW algorithm gives a quality index from 1 to 3 to each boundary layer height candidate. The quality index value is determined based on the gradient magnitude, detected cloud base, and the distance of the local gradient minimum to other gradient minima. A low gradient results in a high quality index; clouds detected in the vicinity of a boundary layer reduces its quality index; and a large distance to other gradient minima results in a high quality index. We select the boundary layer height candidate with the highest quality index as the

ceilometer estimated PBLHT.





Figure 2 shows an example of Vaisala CL31 total attenuated backscatter coefficient profiles and estimated PBLHT from ceilometer data as well as from the ARM PBLHT-SONDE VAP on February 9-10, 2015, at the ARM SGP site. From figure 2a) we can see the presence of several aerosol layers and their evolution with time. Starting at 18:00 LT local time (LT) on

February 9, a residual aerosol layer with a top of ~ 0.7 km AGL was present and its top descended gradually with time, which caused a large variation of the ceilometer estimated PBLHT. This situation represents a challenging scenario to use the aerosol gradient to estimate PBLHT. At ~21:00 LT on February 9, another dense aerosol layer was formed near the surface and started to grow steadily to a height of approximately 0.3 km AGL until ~23:00 LT. After 23:00 LT the residual aerosol layer and the dense surface aerosol layer were separated. The residual aerosol layer was forced to ascend slightly and then disappeared at

~2:00 LT on February 10, probably was advected out of the ceilometer field of view or was entrained into the lower mixed-layer zone. While the dense surface aerosol layer stayed quite stable within the boundary layer until ~9:00 LT and then started to grow quickly from ~0.3 km AGL at 9:00 LT to ~1.2 km AGL at 18:00 LT. As the aerosol layer expanding to a higher altitude, its density decreased as revealed by the decrease of ceilometer backscatter coefficient values after 12:00 LT. Figure 2b) shows that ceilometer-estimated PBLHTs match well with the evolution of aerosol layers shown in Figure 2a), which

demonstrates the advantage of using high-temporal resolution continuous PBLHT data for studying boundary layer structures. PBLHTs from the ARM PBLHT-SONDE VAP at each radiosonde launching time are also plotted in Figure 2 with different colors for different PBLHT estimation methods. Ceilometer-estimated PBLHTs agree well with that from the PBLHT-SONDE VAP at 5:30 and 11:30 LT on February 10 radiosonde launches and PBLHTs from the PBLHT-SONDE VAP have a narrow range. Ceilometer-estimated PBLHTs agree well with the bulk Richardson number method at 23:30 LT on February 9 and

with the Heffter method at 17:30 LT on February 10. PBLHTs from the PBLHT-SONDE VAP span a large range at these two time periods.

To better understand PBLHT estimations from ceilometer data and radiosonde data, Figure 3 shows profiles of ceilometer backscatter coefficient, radiosonde derived potential temperature, and Richardson number that are used to estimate PBLHT in

different methods. Estimated PBLHTs from ceilometer data and the ARM PBLHT-SONDE VAP are also plotted. At 23:30 LT on February 9, the boundary layer is stable as seen in the potential temperature profile. Ceilometer backscatter coefficient profile shows a strong negative gradient, but potential temperature and bulk Richardson number show a strong positive gradient at the height of 0.3 km AGL, which agrees well with PBLHT CEIL and PBLHT Richardson. PBLHT Heffter and PBLHT Liu-Liang are underestimated. At 5:30 LT on February 10, the boundary layer is still stable. PBLHT CEIL, PBLHT Richardson,

and PBLHT Liu-Liang agree well, but PBLHT Heffter is underestimated. At 11:30 LT on February 10, the boundary layer is well-mixed and all PLBHT estimations agree well. At 17:30 LT on February 10, there is a weak stable layer developed near the surface, where the low altitude atmosphere is still well-mixed. This is a typical structure of a residual layer overlaying a weak stable layer. PBLHT CEIL and PBLHT Heffter captured the top of the residual layer, while PBLHT Liu-Liang is underestimated. PBLHT bulk Richardson is quite low, but it might take the top of the weak stable layer as the PBLHT.




It should be noted that PBLHT CEIL performed well for this day. However, it is not uncommon that there are days when aerosol loading is not strong or there are advected aerosol layers that cause trouble for accurate PBLHT estimations from ceilometer measurements. Therefore, for the rest of the sections, we will focus on statistical comparisons of these PBLHT estimations using ARM measurements and the PBLHT-SONDE VAP at different ARM fixed-location observatories and AMF

field campaigns.

## 3 Results and Discussions

As discussed in the preceding section, the performance of PBLHT estimation methods might be impacted by the boundary layer stability and the surface type. Literature suggests that the presence of low-level clouds could also impact PBLHT estimations. Therefore, we separate comparisons of PBLHT CEIL and PBLHT SONDE for different boundary layer regimes

and cloudy and cloud-free conditions. Figure 4 shows occurrence fractions of different boundary layer regimes, cloudy, and cloud-free conditions at ARM fixed-location observatories and AMF deployments. Figure 4a) shows that MAO, NSA, OLI, and AWR are dominated by the SBL regime; while TWP, ASI, SGP, ENA, and COR are dominated by the NRL regime. The CBL regime generally has a small fraction for all the observatories and is negligible for the TWP, MAO, and SGP observatories. Since the CBL and NRL have similar state variables and pollutant profiles, we combine the CBL and NRL

regimes and refer to it as the unstable boundary layer condition, in contrast to the SBL regime, which stands for the stable boundary layer condition. To investigate possible impacts of clouds, comparisons are also separated for conditions with and without the presence of low-level clouds below 4 km AGL (referred to as LLC and LLC-free, correspondently), as detected by ceilometer at the time of radiosonde launching. Figure 4 b) shows that majority of the observatories have an LLC fraction greater than 0.6. Especially, ASI, ENA, NSA, and OLI are largely dominated by LLC.

### 3.1 Low-level Cloud-free Unstable Boundary Layer Conditions

To statistically compare PBLHT CEIL and PBLHT SONDE estimations, Figure 5 shows the correlation coefficients between PBLHT CEIL and PBLHT SONDE estimations with different methods at the nine ARM observatories under LLC-free unstable boundary layer conditions. As a reference, correlation coefficients between PBLHT Heffter and PBLHT Liu-Liang are also plotted ($R_{Heffter-LiuLiang}$). From Figure 5 a), PBLHT CEIL has higher correlation coefficients with PBLHT Liu-Liang ($R_{CEIL-}$

$_{LiuLiang}$) than PBLHT Heffter ($R_{CEIL-Heffter}$) and PBLHT Richardson ($R_{CEIL-Richardson\_p25}$ $and$ $R_{CEIL-Richardson\_p5}$) at all ARM observatories except OLI. One reason that PBLHT Liu-Liang performs well might be because the Liu-Liang method uses different methods and thresholds for different boundary layer regimes and surface types. Sawyer and Li (2013) and Su et al., (2020) suggested that their PBLHT estimations using Micro-Pulse Lidar (MPL) measurements compared better with PBLHT Liu-Liang and preferred to use PBLHT Liu-Liang data to evaluate their PBLHT estimations at the ARM SGP observatory. Su

et al. (2020) shows that their PBLHT estimations with eight years of MPL data using a wavelet method have a correlation





coefficient of 0.61 with PBLHT Liu-Liang under NRL boundary layer conditions at the ARM SGP site, which is slightly higher than the correlation coefficient of 0.54 for our PBLHT CEIL and PBLHT Liu-Liang comparisons. This could be because MPL operates at the wavelength of 532 nm, which is more sensitive to sub-micron aerosol particles; while ceilometer operates at the wavelength of 910 nm, which is less sensitive to sub-micron aerosol particles and might miss thin aerosol layers. PBLHT

CEIL has higher correlation coefficients with PBLHT Heffter than PBLHT Richardson at most ARM observatories. Seibert et al. (2000) suggested that parcel methods using potential temperature profiles are more reliable for PBLHT estimations under convective boundary layer conditions. PBLHT Richardson using $Ri_c$ values of 0.25 and 0.5 do not produce statistically different comparisons with PBLHT CEIL. At different ARM observatories, PBLHT CEIL and PBLHT SONDE comparisons show dramatic differences. Low- and mid-latitude land observatories including MAO, SGP, and COR have higher correlation

coefficients between PBLHT CEIL and PBLHT SONDE than other observatories, indicating surface type impacts on the comparisons. PBLHT Heffter and PBLHT Liu-Liang comparisons also show high correlation coefficients at MAO, SGP, and COR, and weak correlation coefficients at TWP, ASI, and ENA, suggesting that it is still challenging to provide reliable PBLHT estimations at these locations even using radiosonde measurements. It is also noted that correlation coefficients at ASI and AWR show a broad spread, which might be caused by small samples over these two sites. From Figure 4, these two

observatories are either dominated by LLC or under SBL boundary layer conditions.

Although these correlation coefficients show the covariances between PBLHT CEIL and PBLHT SONDE, they do not provide information on absolute differences between PBLHT CEIL and PBLHT SONDE. Kernel density estimates (KDE), which represent the distribution of observations in datasets, are shown in Figure 6 for PBLHT CEIL and PBLHT Liu-Liang under

LLC-free unstable boundary layer conditions at the nine ARM observatories. Since PBLHT CEIL has higher correlation coefficients with PBLHT Liu-Liang ($R_{CEIL-LiuLiang}$) under LLC-free unstable boundary layer conditions, we prefer to show KDE plots for comparisons between PBLHT CEIL and PBLHT Liu-Liang among all PBLHT estimation methods using radiosonde data. Consistent with correlation coefficients in Figure 5, MAO, SGP, and COR, which have high correlation coefficients, also show better comparisons between PBLHT CEIL and PBLHT Liu-Liang. At ASI and ENA, which have low correlation

coefficients, PBLHT CEIL is generally lower than PBLHT Liu-Liang, probably because these observatories are over the ocean and do not have strong aerosol loadings. While at NSA and OLI, PBLHT CEIL is generally higher than PBLHT Liu-Liang, probably because of the presence of transported aerosol layers from low latitudes.

**3.2 Low-level Cloud-free Stable Boundary Layer Conditions**

As being pointed by many previous studies, it is still challenging to obtain reliable PBLHT estimations under stable boundary

layer conditions even using *in situ* radiosonde data (Seibert et al., 2000). Figure 7 shows the correlation coefficients between PBLHT CEIL and PBLHT SONDE estimations with different methods at the nine ARM observatories under LLC-free stable boundary layer conditions. As expected, most correlation coefficients including those of PBLHT Heffter and PBLHT Liu-



Liang are close to zero, and some comparisons are even negatively correlated, suggesting significant differences in PBLHT estimations under stable boundary layer conditions among different methods. $R_{CEIL\text{-}Richardson\_p25}$, $R_{CEIL\text{-}Richardson\_p5}$, and $R_{CEIL\text{-}Heffter}$

are weakly positive at TWP, SGP, COR, and AWR. Su et al. (2020) shows a correlation coefficient of 0.27 for MPL derived PBLHT and PBLHT Liu-Liang under stable boundary layer conditions at the ARM SGP site, compared to a correlation coefficient of -0.03 for PBLHT CEIL and PBLHT Liu-Liang comparison in this study. This could be because the method they used can only estimate PBLHT several hundred meters above the ground due to MPL near-surface 'blind zone', overlap corrections, and the 'dilation' parameter used to conduct the wavelet transform. Correlation coefficients at ASI show a broad

spread, which might be caused by small samples as ASI is dominated by LLC.

Since $R_{CEIL\text{-}Richardson\_p25}$ shows positive values for several ARM observatories, KDE plots for comparisons between PBLHT CEIL and PBLHT Richardson_p25 under LLC-free stable boundary layer conditions are shown in Figure 8. KDE plots for comparisons between PBLHT CEIL and PBLHT Heffter, and PBLHT Liu-Liang, as well as related discussions, are presented

in the Appendix (Figure S1 and S2, separately). From Figure 8, although correlation coefficients are low, absolute differences between PBLHT CEIL and PBLHT Richardson_p25 are not large, as can be seen, that the maximum occurrences of KDE are generally located close to the 1:1 line. This is because PBLHT is low under stable boundary layer conditions. PBLHT CEIL often shows much larger values than PBLHT Richardson_p25. This is expected because PBLHT CEIL tends to pick the top of the residual layer or elevated aerosol layer under stable boundary layer conditions.

## 3.3 Low-level Cloudy Conditions

In the presence of LLC, the ceilometer backscatter coefficient profile shows a sharp gradient at the cloud layer level due to strong attenuation of ceilometer signal by cloud droplets, which could be captured by the ceilometer PBLHT detection algorithms as a PBLHT candidate. Most LLC cloud bases occur at or close to the top of the boundary layer. Indeed, comparisons of PBLHT CEIL and ceilometer LLC cloud base show that in general, they match well, except that sometimes

ceilometer LLC cloud bases are higher than PBLHT CEIL (Figure S3). This is because ceilometer detected clouds could be advected from other locations or are formed from moisture layers that are advected from other locations. Correlation coefficients for PBLTH CEIL and PBLHT SONDE comparisons under LLC unstable boundary layer conditions are shown in Figure 9. Similar to the LLC-free unstable boundary layer conditions, low- and mid-latitude land observatories including MAO, SGP, and COR have higher correlation coefficients between PBLHT CEIL and PBLHT SONDE, while ASI and ENA

have low even negative correlation coefficients. KDE plots for PBLHT CEIL and PBLHT Liu-Liang are shown in Figure 10. Good agreements between PBLHT CEIL and PBLHT Liu-Liang are shown at MAO, SGP, and COR. Under LLC stable boundary layer conditions, LLCs are often decoupled from the boundary layer. Therefore, correlation coefficients between PBLHT CEIL and PBLHT SONDE are low (Figure S4) and PBLHT CEIL is generally higher than PBLHT SONDE (Figure S5).



### 3.4 PBLHT Diurnal Evolution and Seasonal Variations

The diurnal evolution of PBLHT is important to better understand boundary layer processes. One advantage of PBLHT estimations with remote sensing measurements is that they provide continuous PBLHT estimations that can be used to study PBLHT diurnal evolution. To compare PBLHT CEIL and PBLHT SONDE at different times of a day, Figure 11 shows box and whisker plots of PBLHT diurnal cycles and their seasonal variations from PBLHT CEIL, PBLHT Liu-Liang, and PBLHT Richardson_p25 at the ARM SGP observatory. There are clear PBLHT diurnal evolutions at all seasons from all PBLHT estimation methods at the ARM SGP observatory. In general, the boundary layer stays shallow within about 1 km AGL during nighttime and starts to grow at ~ 9:00 am local time, reaches its peak value at late afternoon, and then begins to decay. This is consistent with past studies (Sawyer and Li 2013; Su et al., 2020). Comparing different PBLHT estimation methods, PBLHT CEIL produces overall higher PBLHTs during nighttime when boundary layers are mostly SBL and comparable PBLHTs with PBLHT SONDE during daytime. Especially, after ~20:00 local time, PBLHT CEIL is much higher than PBLHT SONDE because PBLHT CEIL tends to detect the residual layer, while PBLHT SONDE tends to detect the stable boundary layer. PBLHT Liu-Liang produces low PBLHTs during nighttime that are generally within 0.4 km AGL, which are always lower than PBLHT Richardson_p25. This suggests that PBLHT Liu-Liang may need to adjust the thresholds used to derive PBLHT under stale boundary layer conditions. PBLHT Richardson_p25, on the other side, often produces low PBLHTs in the afternoon compared with PBLHT CEIL and PBLHT Liu-Liang, which indicates that the bulk Richardson method is not suitable to provide reliable PBLHT estimations for strong convective boundary layer conditions. For different seasons, Summer has the largest PBLHT diurnal evolution as well as the highest PBLHTs during the afternoon under convective boundary layer conditions, while Winter has the smallest PBLHT diurnal evolution and lowest PBLHTs, mainly because Summer has stronger surface convections than other seasons.

### 4 Summary and Conclusions

Ceilometer provides continuous measurements of aerosol backscatter profiles, which have been widely used to estimate the planetary boundary layer height (PBLHT). Good agreements between ceilometer- and radiosonde-estimated PBLHTs have been reported using limited data from a single location or a short-term campaign. To test the robustness of ceilometer-estimated PBLHT under different atmospheric conditions, we compared ceilometer- and radiosonde-estimated PBLHTs using multiple years of U.S. DOE ARM measurements at six ARM observatories located around the world.

The ARM PBLHT-SONDE Value-added Product (VAP) implements three commonly used methods including the Heffter method, the Liu-Liang method, and the bulk Richardson number method to estimate PBLHT from radiosonde data. The Vaisala CL31 ceilometer at ARM observatories identifies up to three boundary layer height candidates from total backscatter coefficient profile measurements using the enhanced gradient method and assigns a quality index to each candidate. The boundary layer height candidate with the highest quality index is selected as the ceilometer-estimated PBLHT (PBLHT CEIL).



We first compared PBLHT CEIL and PBLHT SONDE estimations for an example day on February 10, 2015, at the ARM Southern Great Plains (SGP) observatory. By examining the ceilometer backscatter coefficient, radiosonde derived potential temperature, and Richardson number profiles, we found that PBLHT CEIL performed well at all the four radiosonde launching times for this day. Then statistical comparisons of PBLHT CEIL and PBLHT SONDE under different atmospheric conditions including stable and unstable boundary layers, low-level cloud-free (LLC-free), and LLC cloudy conditions were performed at different ARM observatories. Under unstable boundary layer conditions, ARM low- and mid-latitude land observatories have higher correlation coefficients and good comparisons between PBLHT CEIL and PBLHT SONDE, while ARM observatories at the ocean surface have weak correlation coefficients. Comparisons between different methods used in PBLHT SONDE show similar features, indicating that it is still quite challenging to provide reliable PBLHT estimations over the ocean surface. Under stable boundary layer conditions, however, most correlation coefficients including those for comparisons between different methods used in PBLHT SONDE are close to zero or even negative, except those comparisons between PBLHT CEIL and PBLHT Richardson show agreement at several ARM observatories. This suggests that it is still challenging to obtain reliable PBLHT estimations under stable boundary layer conditions, even using *in situ* radiosonde data, and the Richardson method is more suitable for estimating PBLHT for these conditions. Overall, the presence of LLC has little impact on the comparisons between PBLHT CEIL and PBLHT SONDE. We further compared PBLHT CEIL and PBLHT SONDE at different times of a day by examining the PBLHT diurnal evolution at the ARM SGP observatory. PBLHT CEIL produces overall higher PBLHTs during nighttime when boundary layers are mostly under stable conditions and comparable PBLHTs with PBLHT SONDE during daytime.

Our statistical comparisons between PBLHT CEIL and PBLHT SONDE at the ARM SGP observatory are similar to past studies that compared Micropulse-lidar (MPL) estimated- with radiosonde-estimated PBLHTs using multiple years of data (Sawyer and Li 2013; Su et al., 2020), but are not as good as those comparisons using limited data from a single location or a short-term campaign (Haeffelin et al., 2012; Haman et al., 2012). The main reason is that in those studies PBLHTs were manually selected from ceilometer backscatter local gradient minimum, while we used an automatic selection method that may fail to pick up the correct PBLHT candidate under stable boundary layer conditions or when a strong elevated aerosol layer is detected. Therefore, advanced PBLHT estimation methods are still needed to improve PBLHT estimations from both ceilometer and radiosonde data. Comparisons of different PBLHT estimation methods could help provide an uncertainty range for PBLHT. On the other hand, the residual layer top detected by ceilometer during nighttime could provide useful information to quantitatively study the impacts of residual layer on the development of boundary layer.

**Data availability**



The ceilometer backscatter and boundary layer height data and ARM PBLHT VAP data used in this study can be downloaded from the ARM data archive site: https://www.archive.arm.gov/discovery/.

**Author contributions**

DZ analyzed the data and wrote the manuscript draft; JC and VM reviewed and edited the manuscript.


**Competing interests**

The authors declare that they have no conflict of interest.

**Acknowledgements**

We thank the personnel responsible for the challenging operations and data collection for ARM observatories at various locations. Data were obtained from the Atmospheric Radiation Measurement (ARM) user facility, a U.S. Department of Energy (DOE) office of science user facility managed by the Biological and Environmental Research (BER) program. This research was supported by the DOE ARM program. The authors also want to thank the anonymous reviewers for their helpful comments in improving the manuscript.

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

**Table 1 ARM observatory site specifics and number of launched radiosondes during the time periods examined in this study**

| Observatory | Elevation ASL (m) | Surface characteristics | Climate/ cloud Regime | Period | Number of radiosonde releases |
|---|---|---|---|---|---|
| **SGP** | 314 | Land | A wide variety of cloud types | 2012/06/08- 2021/08/07 | 12416 |
| **NSA** | 8 | Tundra/ice | Seasonal ice cover, polar mixed-phase clouds | 2013/08/05- 2021/08/05 | 7084 |





| ENA | 30 | Ocean | Marine stratus/ stratocumulus | 2013/09/29- 2021/08/07 | 6074 |
|---|---|---|---|---|---|
| TWP-Darwin | 30 | Ocean | Deep tropical convection; cirrus clouds | 2013/11/14- 2015/01/03 | 1624 |
| MAO | 50 | Land | Deep tropical convection | 2013/12/17- 2015/12/01 | 2888 |
| ASI | 76 | Ocean | Martine stratocumulus | 2016/05/03- 2017/10/30 | 2270 |
| COR | 1141 | Land | Shallow and deep convective clouds | 2018/12/28- 2019/04/29 | 487 |
| AWR | 10 | Tundra/ice | Ice surface, polar mixed-phase clouds | 2015/11/30- 2017/01/02 | 785 |
| OLI | 2 | Tundra/ice | Seasonal ice cover, polar mixed-phase clouds | 2014/11/12- 2021/06/13 | 3921 |





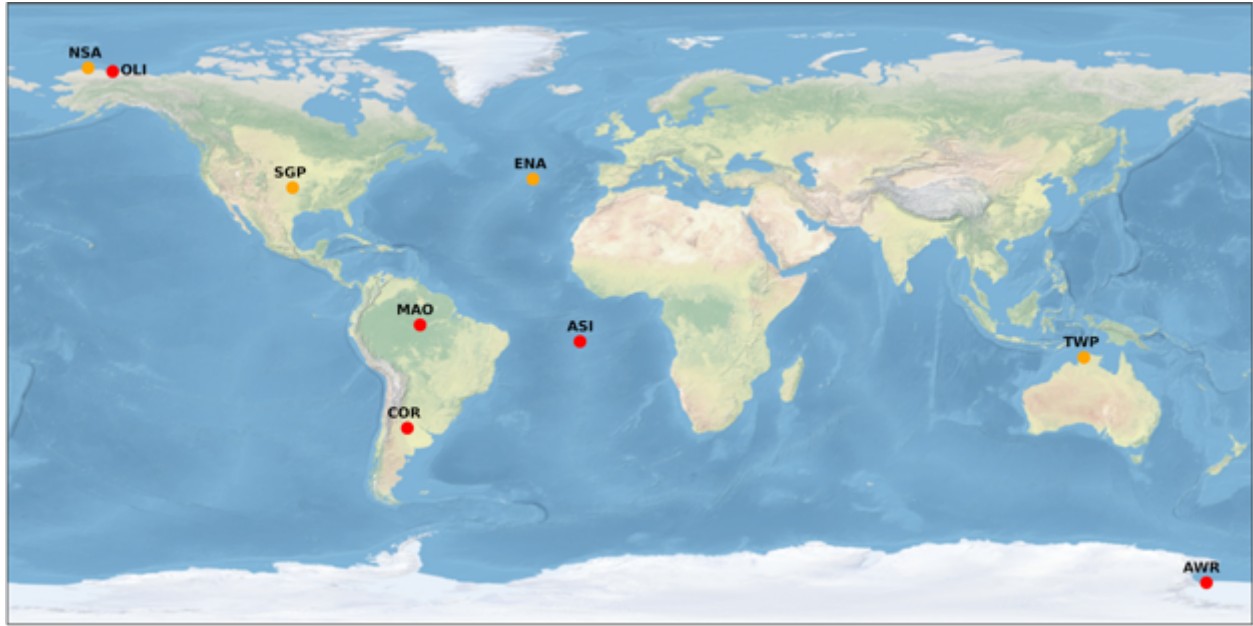

**Figure 1: The geographical locations of ARM fixed-location observatories (orange circles) and AMF field campaign deployments (red circles).**





**Figure 2: An example of estimated PBLHTs from ceilometer and the PBLHT-SONDE VAP on February 9-10, 2015,**

**at the ARM SGP site. a) time-height cross section of ceilometer total attenuated backscatter coefficient; b) estimated**

**PBLHTs from ceilometer measurements (labeled as 'CEIL') and the PBLHT-SONDE VAP including the Heffter**

**(labeled as 'Heffter'), Liu and Liang (labeled as 'Liu-Liang'), and bulk Richardson number (using $Ri_c$ of 0.25 is**

**labeled as 'Richardson_p25' and using $Ri_c$ of 0.5 is labeled as 'Richardson_p5') methods.**






Figure 3: Ceilometer backscatter coefficient, radiosonde derived potential temperature and Richardson number profiles at the radiosonde launching time of a) 23:30 Local Time (LT) on February 9; b) 5:30 LT on February 10; c) 11:30 LT on February 10; and d) 17:30 LT on February 10. Estimated PBLHTs from ceilometer data and the ARM

PBLHT-SONDE VAP are also plotted as triangle signs with different colors for different methods. The color for each method is the same as in Figure 1b.

**Figure 4: Occurrence fractions of a) different boundary layer regimes; b) low-level cloudy and cloud-free conditions**

**at the nine ARM observatories during the time periods examined in this study. SBL, NRL, and CBL stand for stable**

**boundary layer, neutral residual layer, and convective boundary layer correspondently in plot a). LLC-free and LLC**

**stand for low-level cloud free and low-level cloudy conditions correspondently in plot b).**





**Figure 5: a) correlation coefficients between PBLHT CEIL and PBLHT Heffter ($R_{CEIL-Heffter}$), PBLHT Liu-Liang ($R_{CEIL-LiuLiang}$), PBLHT Richardson ($R_{CEIL-Richardson\_p25}$ and $R_{CEIL-Richardson\_p5}$). Correlation coefficients between PBLHT Heffter and PBLHT Liu-Liang are also plotted ($R_{Heffter-LiuLiang}$); b) correspondent radiosonde amounts at ARM observatories under the LLC-free unstable boundary layer conditions.**






**Figure 6: Kernel distribution estimate for PBLHT CEIL and PBLHT Liu-Liang under LLC-free unstable boundary layer conditions at the nine ARM observatories. Blue dashed lines are the 1:1 line. *R* is the correlation coefficient and *n* is the sample number.**






**Figure 7: similar to Figure 5, but for LLC-free stable boundary layer conditions.**

**Figure 8: similar to Figure 6, but for PBLHT CEIL and PBLHT Richardson_p25 under LLC-free stable boundary layer conditions.**



**Figure 9: similar to Figure 5, but for LLC unstable boundary layer conditions.**






Figure 10: similar to Figure 6, but for LLC unstable boundary layer conditions.



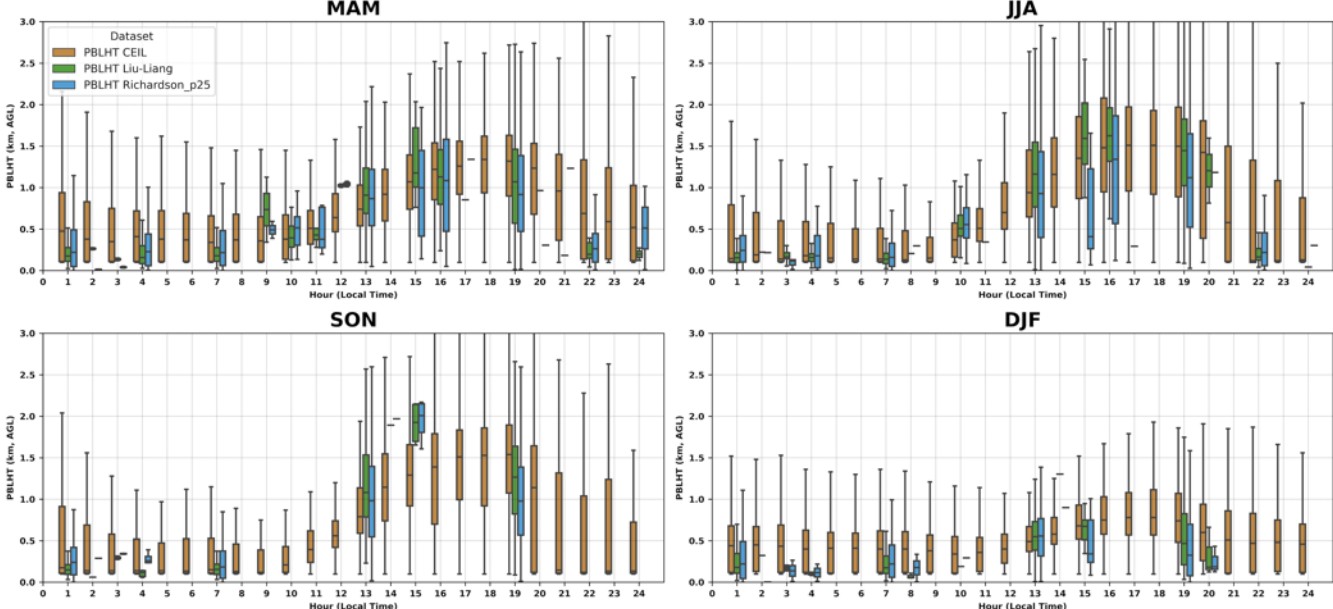


**Figure 11: PBLHT diurnal cycles and their seasonal variations from PBLHT CEIL, PBLHT Liu-Liang, and PBLHT Richardson_p25 at the ARM SGP observatory. MAM (March-April-May) represents the Spring season, JJA (Jun-July-August) for Summer, SON (September-October-November) for Fall, and DJF (December-January-February) for Winter. Horizontal bars, boxes and whiskers represent the median, interquartile range, and range of the data.**
