# Peer review of "Comparison of Planetary Boundary Layer Height from Ceilometer with ARM Radiosonde Data"

_Atmospheric Measurement Techniques, 2021_

## Author Response (AR1)

*Reply to Reviewer #1's comments*

- *The author aimed to compare the ceilometer- and radiosonde-estimated PBLHTs under stale, unstable and RL, cloudy and cloud-free conditions. But what is the stability parameter used and how is the RL defined in this study? How the cloudy and cloud-free condition is defined? It should be explained.*

We pointed out in line 145 that 'The Liu-Liang method classifies the boundary layer regime as CBL, SBL, or NRL by comparing $\theta_5 - \theta_2$ with a stability threshold $\delta_s$'. We added another sentence at line 146 'For CBL, $\theta_5 - \theta_2 < -\delta_s$; for SBL, $\theta_5 - \theta_2 > +\delta_s$; and for NRL, $-\delta_s < \theta_5 - \theta_2 < +\delta_s$' to provide more details about how the stability regime is determined.

We pointed out in line 179 that 'The Vaisala CL31 ceilometer detects up to three cloud layers simultaneously and measures vertical visibility'. To make it more clear, we added a sentence in line 180 'Ceilometer cloud detections are used to distinguish cloudy and cloud-free conditions'.

- *The observation data used in this paper include both over land and ocean. However, what is the difference between the accuracy of PBLH estimation over land and ocean? It is suggested to be explained in the manuscript.*

We agree with the reviewer that when evaluating the retrieved variables, it is important to examine the retrieval bias, accuracy/differences between the retrievals and the ground truth. However, a great challenging for PBLHT estimations is that there is no ground truth to evaluate with. We pointed this out in the line 221. It is difficult to obtain the overall accuracy of the two ways of estimating PBLHTs. We believe that good comparisons between ceilometer- and radiosonde-estimated PBLHTs generally indicate more reliable PBLHT estimations.

- *In Figure 2, what is the reason for the great difference in PBLH retrieved by different methods at 18:00 LT? According to the attenuated backscatter coefficient, it is well mixed within the PBL, generally, the uncertainty of PBLH retrieving should be relatively small under this condition?*

We agree with the reviewer that for well mixed PBL, the uncertainty of PBLHT retrieving should be relatively small. The PBL structure at 18:00 LT is more complicated. In lines 227-229 we pointed out that 'At 17:30 LT on February 10, there is a weak stable layer developed near the surface, where the low altitude atmosphere is still well-mixed. This is a typical structure of a residual layer overlaying a weak stable layer. PBLHT CEIL and PBLHT Heffter captured the top of the residual layer, while PBLHT Liu-Liang is underestimated. PBLHT bulk Richardson is quite low, because it takes the top of the weak stable layer as the PBLHT'.

- *In Figure 3, The profiles of backscatter and Richardson number is incomplete, which will lead us to doubt the rationality of the data. In addition, what are the reasons for the difference of PBLH retrieved by different methods? Because the defect of the method or the structures of the PBL? should be explained.*

The Vaisala CL31 ceilometer has a field-of-view of 0.83 *mrad* and receives considerable background signals when pointing vertically. Therefore, subtracting background signals during the post-processing procedure leads to noisy ceilometer backscatter profiles above PBL when the atmosphere is free of clouds or aerosol layers. Early studies show that CL31 ceilometer is capable of detecting aerosol layers and can be used to estimate PBLHT (Münkel et al., 2007).

The Bulk Richardson number increases dramatically above PBL and is out of the x-axis range in Figure 3. We pointed out in line 154 that 'The bulk Richardson number $Ri$ represents the ratio of thermally produced turbulence to that generated by vertical wind shear. Since wind shear produced turbulence is greatly reduced above the top of atmospheric boundary layer, $Ri$ increases dramatically at the top of SBL.'

As for the reasons of the difference of PBLHT retrieved by different methods, we believe that it is because of the limitation of the measurements, the defect of the methods, and the complicated structures of the PBL. (1) temperature, humidity, and aerosol intensity measurements only indirectly reflect PBL structures. Direct measurements of PBL turbulence structures with high temporal and vertical resolutions are not available; (2) the retrieval methods are often based on empirical relations, which might not be applicable to certain complicated PBL structures; (3) it is still challenging to obtain reliable PBLHT estimations under stable boundary layer conditions. In this study, we show that 'under unstable boundary layer conditions, ARM low- and mid-latitude land observatories have higher correlation coefficients and good comparisons between PBLHT CEIL and PBLHT SONDE. ARM observatories at the ocean surface and under stable boundary layer conditions have weak correlation coefficients between PBLHT CEIL and PBLHT SONDE.'

Reference:

Münkel, C., Roininen, R.: Automatic Monitoring of Boundary Layer Structures with Ceilometer. vol. 184 Vaisala News., 2010.

*Reply to Reviewer #2's comments*

*Main comment:*
*This is an interesting paper that has an element of novelty: it presents a comparison between ceilometer and radiosonde data for a range of PBL sites and regimes. I generally think that the paper is worth publishing although it needs major improvements.*

**We thank the reviewer for these constructive suggestions and comments. We carefully revised the manuscript according to the reviewer's comments.**

*Specific comments:*

*You show correlation coefficients, but I don't see any calculations of and the discussion on the biases between the tested methods. Can you please add it (if not to the paper, then to the Supplement)? This is an important part that should complete the whole picture.*

*What is the significance of your correlation coefficients? You calculate those values but don't seem to comment on their significance. To me it looks like only convective PBL can be relatively well probed with the two instruments but I couldn't find anything about the overall accuracy/differences between the two ways of calculating it.*

**We thank the reviewer for these two very constructive comments. We agree with the reviewer that when evaluating the retrieved variables, it is important to examine the retrieval bias, accuracy/differences between the retrievals and the ground truth. However, a great challenging for PBLHT estimations is that there is no ground truth to evaluate with. We pointed this out in the line 221. It is difficult to obtain the overall accuracy of the two ways of estimating PBLHTs. We believe that good comparisons between ceilometer- and radiosonde-estimated PBLHTs generally indicate more reliable PBLHT estimations.**

*You could shorten your PBLHT acronym to PBLH and still be well understood.*

**We thank the reviewer for the suggestion. In literature both PBLHT and PBLH have been used (Huang et al., 2011; Nelson et al., 2021). We prefer PBLHT in the manuscript to keep consistent with the ARM PBLHT-VAP.**

**References:**

**Huang, M., et al.: Multi-scale modeling study of the source contributions to near-surface ozone and sulfur oxides levels over California during the ARCTAS-CARB period, Atmos. Chem. Phys., 11, 3173–3194, https://doi.org/10.5194/acp-11-3173-2011, 2011.**

**Nelson, K. J., Xie, F., Ao, C. O., & Oyola-Merced, M. I. (2021). Diurnal Variation of the Planetary Boundary Layer Height Observed from GNSS Radio Occultation and Radiosonde Soundings over the Southern Great Plains,** *Journal of Atmospheric and Oceanic Technology* **(published online ahead of print 2021).**

*Abstract:*
*L7: How can a parameter influence atmospheric processes? Processes are controlled by the physics, not by PBLHT. Maybe you meant that their representation in climate models depends on PBLHT?*

**We agree with the reviewer that processes are controlled by the physics. We deleted this sentence.**

*You mention PBL types (stable vs unstable, cloud free and cloudy conditions. More information is needed on latitudes (mid-lat vs tropics?) and types of the surface (maritime vs continental).*

**As suggested, we added a sentence of "which cover from Tropics to Polar regions and over both ocean and land surfaces" in the line 13.**

*"Under unstable boundary conditions" – you can simply say "For convective boundary layers"*

**We thank the reviewer for the suggestion. As shown in Figure 4 and discussed in lines between 403-407, "The CBL regime generally has a small fraction for all the observatories and is negligible for the TWP, MAO, and SGP observatories. Since the CBL and NRL have similar state variables and pollutant profiles, we combine the CBL and NRL regimes and refer to it as the unstable boundary layer condition, in contrast to the SBL regime, which stands for the stable boundary layer condition." Therefore, we used "unstable boundary layer conditions" in stead of calling it "convective boundary layers".**

Introduction:
L24: and moisture

**We added this in the sentence.**

*L27-30: Unclear: is the depth a parameter that determines the structure of the lowest few km? The depth is the result rather than a cause of the PBL processes.*

**We agree with the reviewer and changed the word "determines" to "characterizes".**

*L33-35: I think a different classification is more common: stable, unstable, neutral PBLs, depending on their mean stratification. This is however not what Liu-Liang method is based on. Generally, that classification is somewhat simplified and may not be appropriate for transient cases, which should be mentioned.*

**We agree with the reviewer that the classification using Liu-Liang method is simplified. We added a sentence in lines between 58-61: "It should be noted that atmospheric boundary layer stability ranges from very stable to strongly unstable. Classification of atmospheric boundary layer stability into these three major regimes is simplified and may not be appropriate for transient atmospheric conditions (Mahrt 1999)."**

*It may be helpful to clarify that Liu-Liang method is simply a temperature gradient method.*

**As suggested, we added a sentence "For example, both Heffter (1980) and Liu and Liang (2010) use potential temperature gradient as a key parameter to determine the PBLHT for CBL and NRL regimes." in lines between 69 and 71.**

*L36: for CBL it is both convection and turbulence that cause strong mixing across the PBL*

**We thank the reviewer for the comment. We added "convection" in the sentence.**

*L37: for shallow CBLs there should be some temperature inversion at the top, but for deeper CBLs (e.g. in the tropics) it is more difficult to determine the top of the PBL*

**We thank the reviewer for pointing out this situation. We added a sentence "For deep CBLs such as in the Tropics, however, it might be difficult to determine the top of the boundary layer using the potential temperature inversion (Kepert et al., 2016)" and a reference in lines between 52 and 54.**

*Mention briefly about different definitions of PBLHT used in atmospheric models and observational studies: maximum Richardson number, temperature inversion, moisture gradient, minimum refractivity gradient.*

**We thank the reviewer for the great suggestion. We added the following sentence to talk about PBLHT determinations in atmospheric models in line 63-65.**

**The PBLHT in atmospheric models is usually calculated by using either diagnostic equations that take surface fluxes and the initial temperature profile as inputs or by using the Richardson number profile to find the first level where the Richardson number exceeds a critical value (Seibert et al., 2000).**

*L127: what does theta with two dots above mean?*

**It should be one dot above theta, which represents the potential temperature gradient. We corrected this typo.**

What is the vertical resolution of the soundings? What is the accuracy of wind and temperature measurements and thus the overall accuracy of the methods used?

We mentioned in line 109-110 that "SONDE … with a 1-second resolution". To make it clearer to readers, we now added in line 165-166: "corresponding to vertical height resolutions of several meters to more than 10 meters depending on the atmospheric dynamic environment."

We also added the accuracies of wind and temperature measurements in line 165: "The accuracies of radiosonde measured temperature and wind speed are 0.2 °C and 0.2 m/2, respectively".

As we pointed out in line 221 that there is not ground truth for PBLHT determinations, it is difficult to have an overall accuracy of the methods used in PBLHT-SONDE VAP.

*L136: Explain why Ri dramatically increase at the top of SBL.*

We added a sentence in the line 201: "Since wind shear produced turbulence is greatly reduced above the top of atmospheric boundary layer,".

Eq. 1: That equation is different from a typical one for bulk Richardson number including temperature and velocity gradients. I think this is because you calculate mean properties in the entire layer between z=0 and PBLH, which should be clarified in the text. I am skeptical about using such a bulk method for determining PBL height. The thicker the layer, the more risk that you omit important turbulence activity between 0 and z. That is why maximum Richardson number method can be more beneficial: instead of looking at one thick layer we can look at a number of thin layers for which Ri is calculated.
Does your Richardson number method really use Eq. 1 or looks into different layers within PBL?

We thank the reviewer for pointing out an alternative way to use the maximum Richardson number method for improving PBLHT estimations with radiosonde data. We use PBLHTs estimated with the bulk Richardson method directly from the ARM PBLHT-SONDE VAP (Sivaraman et al., 2013), which implements the equation (7) in Sørensen et al. (1998). The bulk Richardson number is calculated at each height, so it is not the mean property in the entire layer between z=0 and PBLHT. As we mentioned, the goal of the study focuses on investigating the robustness of ceilometer-estimated PBLHTs. Improving PBLHT estimations from radiosonde data using the maximum Richardson number method is out of the scope of this study.

References:

Sivaraman, C., McFarlane, S., Chapman, E., Jensen, M., Toto, T., Liu, S., and Fischer, M.: Planetary boundary layer (PBL) height value added product (VAP): Radiosonde retrievals, U.S. Department of Energy Rep. DOE/SC-ARM-TR-132, 36 pp., https://www.arm.gov/publications/tech_reports/doe-sc-arm-tr-132.pdf, 2013.

**Sørensen, J.H., Rasmussen, A., Ellermann, T. and Lyck, E.: Mesoscale Influence on Long-range Transport – Evidence From ETEX Modeling and Observations,** *Atmospheric Environment,* **32(24): 4207–4217, https://doi.org/10.1007/978-1-4615-4153-0_27, 1998.**

*3.1 LLC-free unstable boundary layer conditions – I suggest to make those section titles more self-descriptive, for instance:*
*LLC-free unstable boundary layer conditions -> Cloud-free unstable PBL (or similar)*
*LLC-free stable boundary layer condition -> Cloud-free stable PBL (or similar)*

**As suggested, we changed those section titles to be more self-descriptive.**

*L315: typo: Richardons -> Richardson*

**We corrected the typo.**

*L361: You claim that "machine learning techniques have the potential to greatly improve PBLHT estimates" – but you didn't prove it in this paper, so this claim has no foundation and should be removed. The same sentence is present in the abstract, and it is totally unjustified as far as I can see.*

**We deleted discussions that are related to machine learning techniques in both the abstract and summary sections.**

*Figures;*
*Fig 3: What are the units of backscatter?*

**The units of ceilometer backscatter coefficient is $sr^{-1}m^{-1}$. To reduce the data storage size and be easier shown in the figure, the units of ceilometer backscatter coefficient was converted to (1/(sr*km*10000)). To be consistent with the standard units, we now converted it back to $sr^{-1}m^{-1}$ in both Figure 2 and Figure 3.**

*Fig 4: Is it a climatology? Or just a selected time period? I don't understand it.*
*Explain what fraction means here (fractional occurrence?).*

**Figure 4 shows the occurrence fractions of boundary layer regimes and low-level cloud-free/cloudy conditions during the selected time period. Fraction is the occurrence fraction. We added the information in the figure and the caption.**

*Fig 11: explain the range of the boxes and whiskers in the caption.*

**We added a sentence "Horizontal bars, boxes and whiskers represent the median, interquartile range and range of the data" to explain the range of the boxes and whiskers in the caption.**

*You could cite this paper:*

*1. Fritz, A. M., Lapo, K., Freundorfer, A., Linhardt, T., & Thomas, C. K. (2021). Revealing the morning transition in the mountain boundary layer using fiber-optic distributed temperature sensing. Geophysical Research Letters, 48, e2020GL092238. https://doi.org/10.1029/2020GL092238*

*(they show how important it is to measure PBL transitions with high spatio-temporal resolutions and suggest using a temperature gradient method)*

**We appreciate the reviewer's suggestion. We agree that this paper highlights the need for high spatial-temporal PBLHT estimations. We added a sentence "Observing atmospheric boundary layer transitions with high temporal-spatial resolutions is required to investigate atmospheric thermodynamic processes (Fritz et al., 2021)." and the reference to this paper in lines between 73-74.**

---

## Referee Report (RR1)

**General comment:**

My general impression is that the new version of the manuscript reads better. However, it still needs some improvements before it can be accepted for publication. Although I previously suggested some changes, it seems that the authors need a stronger and more explicit supervision. I found a couple major problems in this version that I already pointed out in my previous review and the authors didn't seem to address them satisfactorily.

**Specific comments:**

1. In my second review, I recommended that the authors read the paper carefully again and check every single sentence for consistency. The authors' response was:
   "**We carefully revised the manuscript according to the reviewer's comments.**"
   When I started reading the new version of the manuscript, the first thing that struck me was the abstract, in which you say:"… at **three** ARM fixed-location atmospheric observatories and from **three** ARM mobile observatories deployed around the world for various field campaigns, which cover from Tropics to Polar regions and over both ocean and land surfaces." and that information is then repeated in the Introduction.

   According to the above, you have six sources of data in total. Just to remind you one of the points from my previous review:
   *L487: "at six ARM observatories located around the world" – This is confusing. Once you say nine (Tab. 1, Fig. 1), then you say six. Again, I strongly encourage the authors to read their paper very carefully and check every single sentence for consistency.*
   **Response: We thank the reviewer for pointing out the typo. The reviewer is correct that it should be nine.**

   Clearly, it was an error not a typo and that error was then repeated in the abstract and the Introduction and is still there. It is difficult for me to understand what the authors mean by "we carefully revised the manuscript" since I find the same simple errors again.

   "which cover from Tropics to Polar regions" – cover what? Use lowercase.

   I suggest to remove the number of stations from the abstract and describe them in a more general way, for instance as a data set from different climate zones probing a variety of PBL regimes.
2. Title: should it be "with" or "and"? The current form suggests that radiosonde data is good and ceilometer data needs to be evaluated.

3. L30: Not true. PBL height does not characterize the structure of the lowest few kilometers of the atmosphere by any means. It only indicates where the top of that PBL structure is located.

4. L36-37: "by an inversion layer of potential temperature" – this sentence is unclear. Is it a temperature inversion? Why "inversion layer"? For convective PBLs potential temperature actually does not have an inversion at the top as its stratification changes from neutral to stable, right? Did you mean real temperature, for which temperature inversion can make more sense?

5. There are at least 9 different methods of estimating PBL height. Von Engeln and Teixeira (2013) mention many of them. In large-eddy simulations, it is common to apply gradient methods (for temperature or moisture) or indeed look at turbulence properties. Some examples worth citing:

   Bopape, M.-J.M.; Plant, R.S.; Coceal, O. Resolution Dependence of Turbulent Structures in Convective Boundary Layer Simulations. *Atmosphere* **2020**, *11*, 986. https://doi.org/10.3390/atmos11090986

   J. Kurowski, M., P. Malinowski, S. and W. Grabowski, W. (2009), A numerical investigation of entrainment and transport within a stratocumulus-topped boundary layer. Q.J.R. Meteorol. Soc., 135: 77-92. https://doi.org/10.1002/qj.354

6. L84: ", the robustness…" – this should either be a new sentence or you should add "and" in between

7. L87: "long term" – this is too vague and means different things for different people; be more specific: multi-year?
8. L97: "low-level cloud-free" – what are low level cloud free conditions here? What does that low level refer to? Does it mean there are clouds at some upper levels?
9. L131-132: "three commonly used methods developed by Heffter (1980; referred to as the Heffter method hereafter)" – does it mean you name the three methods as the Hefter method?
10. That description of 3 methods is still slightly chaotic.
    Please summarize the text between L130 and L155 with an additional table: Method name, algorithm (e.g., Richardson number, potential temperature gradient), reference, etc. which should help understand your methodology.

11. L145: when you say between fifth and second levels of sounding data it means nothing to the reader. Explain what levels 5 and 2 mean. Is it at a fixed height which is the same at all stations under all conditions?
    In principle, if you use a threshold for potential temperature difference, it means you assume that there must exist a minimum gradient. So that method also relies on temperature gradient.
12. L160: We need to discuss this definition of Ri one more time. According to your description: "Ri at a given altitude can be calculated from sounding data"

$$Ri_b = \left(\frac{gz}{\theta_{v0}}\right)\left(\frac{\theta_{vz}-\theta_{v0}}{u_z^2 + v_z^2}\right)$$

As I explained in my previous reviews, this particular definition refers to the entire layer from 0 (surface) up to some height z. This is not Ri at a given altitude and you can only do it for a layer of finite height. If you want to look at a layer of thickness dz, then it should rather be:

$$R_B = \frac{(g/T_v)\Delta\theta_v\Delta z}{(\Delta U)^2 + (\Delta V)^2}$$

Because your definition replaces all the gradients with the values at the height z, it implicitly assumes that those values diminish at z=0 (which is true for u and v, but not for theta that uses a difference between level 0 and z in your equation. How exactly is Ri calculated in your data?

If you still claim that Ri in your equation (1) is given at the height z, then we have a problem with understanding basic equations used in your study. I tried to explain it in my previous reviews, but was so far unsuccessful.

It is critically important that you understand and explain the way you use Ri in your study correctly as it is the foundation of your analysis.

13. L265: "because Liu-Liang method uses different methods" – confusing
14. Fig. 11: Make axes labels and ticks larger.

---

## Author Response (AR2)

*Reply to Reviewer #1's comments*

*General comment:*
*The new version of the manuscript looks better; however, I still see that the quality of the study needs to be improved. I have a serious problem with understanding your definition and use of the Richardson number. If your explanation is true (which I believe you confirmed in the responses), then I think your methodology is used incorrectly, which may invalid your investigation.*

*I strongly recommend that ALL the authors read the paper very carefully again. Please be very critical about every sentence you wrote, and ask yourselves if each of those sentence makes sense. It would also be good to check the final version of the text for errors and confusing formulations. Please see my specific comments below showing how difficult it was for me to understand some parts of the text.*

**Response: We thank the reviewer for these constructive suggestions and comments. We carefully revised the manuscript according to the reviewer's comments.**

*Specific comments:*
*Title: "Comparisons" - I would suggest to change it to a more standard 'Comparison' as your study is a comparison of different approaches to estimating PBLH.*

**Response: We changed the title using 'Comparison' as suggested.**

*Abstract:*
*There is no need to use 'stable atmospheric boundary layer conditions' or 'unstable atmospheric boundary layer conditions' so many times in one paragraph. Once you introduced it, a shorter version ('stable conditions' or 'unstable conditions') can be used.*

**Response: We thank the reviewer for the helpful suggestion on improving the readability of the manuscript. We used a shorter version after 'stable atmospheric boundary layer conditions' or 'unstable atmospheric boundary layer conditions' is introduced in the abstract.**

*L29: Still confusing because PBLHT as a parameter doesn't have any impact on air quality or climate. No parameter has any impact on those elements of the Earth system. I am not sure what the authors are trying to say here. Please consider reformulating it and think about the reader that may not understand everything.*

**Response: We thank the reviewer for pointing out the confusing sentence. We reformulated the sentence in the manuscript as following:**

**The structure and the depth of the planetary boundary layer play a critical role in near surface air quality, land-atmosphere interactions, and a wide range of atmospheric processes such as cloud formation and evolution, aerosol mixing and transport, and aerosol-cloud interactions (Seinfeld et al., 2006; Konor et al., 2009; Lemone et al., 2018). The height of the planetary boundary layer (PBLHT) is a key parameter that characterizes the structure of the lowest few kilometers of the atmosphere.**

*Tropics -> tropics*

**Response: We corrected it as suggested.**

*L54: 'Under the SBL condition, turbulence tends to be suppressed by the statically stable air above it and occurs only sporadically' - What do you mean by 'above it'? Above the boundary layer? But then it doesn't make much sense. This sentence is confusing and needs some clarification. Do you want to involve everything that happens above the PBL?*

**Response: We thank the reviewer for pointing the confusing sentence. We rewrote this sentence in the manuscript in lines 60-62 as following:**

**"The SBL is commonly formed during nighttime by surface radiative cooling or when warm air is advected over a cool surface. Under the SBL condition, virtual potential temperature increases with altitude in the boundary layer. Turbulence tends to be suppressed and occurs sporadically."**

*L62: 'The PBLHT in atmospheric models is usually calculated by using either diagnostic equations that take surface fluxes and the initial temperature profile as inputs ' – This is a very confusing sentence. It wrongly suggests that most of the models (I'm sure you know that atmospheric models include large-eddy simulation, regional, and global circulation models) calculate PBLHT this way, but this is not true. Why only initial conditions? How can PBLHT vary if you use initial conditions? PBLHT may be calculated in many ways, not necessarily by using surface fluxes. In one of the next sentences, you suggest that those methods are only applicable to observational data, which is questionable (once you have some data, you can apply any method you want). Please reformulate this paragraph. You may need to do some research on those methods before you update it.*

**Response: We thank the reviewer for pointing out the confusing sentence. We rewrote the sentence in the manuscript in lines 70 -73 as following:**

**"The PBLHT in numerical weather prediction and climate models is usually calculated by using the Richardson number profile to find the first level where the Richardson number exceeds a critical value and in large-eddy simulation models by using turbulence kinetic energy or eddy diffusivity thresholds (Seibert et al., 2000; Noh et al., 2002; Seidel et al., 2012)."**

**As for the confusing sentences about PBLHT estimation methods for observations, we did not mean that these methods can't be applied to model simulations. To avoid confusion, we deleted the sentence 'On the other hand' in the manuscript.**

*L73: 'Observing atmospheric boundary layer transitions with high temporal-spatial resolutions is required to investigate atmospheric thermodynamic processes (Fritz et al., 2021).' – Another confusing sentence. What thermodynamic processes do you have in mind? That sentence suggests all of them. I think the message from the Fritz et al paper is different as they show that there are many important details that cannot be measured when using instruments with coarser resolution than their DTS, in addition to the fact that their method of determining PBLH based on the temperature gradient is quite reliable.*

**Response: We thank the reviewer for pointing out the confusing sentence. Since this paragraph focuses on PBLHT, we rewrote the sentence to reflect the need for studying PBLHT evolutions using high temporal-spatial observations in lines 80-82 as following:**

**"Observing atmospheric boundary layer transitions with high temporal-spatial resolutions is required to investigate the evolution of PBLHT, which will help to improve its representations in models (Su et al., 2020, Fritz et al., 2021)."**

**We agree with the reviewer that 'Fritz et al paper is different as they show that there are many important details that cannot be measured when using instruments with coarser resolution than their DTS'. In the paragraph, we wanted to emphasize that observations with higher temporal resolutions than ARM radiosonde launching are needed to study the evolution of PBLHT. To make this more clear, we added another reference to Su et al. (2020).**

*L115: 'limited cases' suggests a problem with those cases. Did you mean limited number of cases?*

**Response: We changed it to 'limited number of cases' in the manuscript.**

*L155: found from -> found in (or found from … to… )*

**Response: We thank the reviewer for pointing out the grammar issue. We changed it to 'found in' in the manuscript.**

*L162: Each ARM observatory generally launches four times a day (except twice daily at NSA and OLI) the balloon-borne sounding system (SONDE) at roughly 5:30, 11:30, 17:30, and 23:30 Universal Time Coordinate (UTC) (5:30 and 17:30 UTC at NSA 165 and 17:30 and 23:30 UTC at OLI). – This sentence is confusing. First you say 'each', then you say not each. Please be as precise as possible. You can say something like 'The balloon-borne sounding system is launched four times a day (at 5:30, 11:30, 17:30, and 23:30 UTC) at most of the sites, except twice a day at NSA (5:30 and 17:30 UTC) and OLI (17:30 and 23:30 UTC).*

*Are different UTC times chosen to match similar local times and facilitate morning and afternoon observations of PBL? If yes, please mention that.*

**Response: We thank the reviewer for pointing out the sentence structure issue and suggesting a better sentence structure. We rewrote the sentence based on the reviewer's suggestion.**

**At most ARM sites, SONDE launches are picked at different synoptic time (00Z, 06Z, 12Z, and 18Z) depending on the scientific justification, staffing, etc. For OLI, the SONDE launch times were chosen because it was when the observers were on site -7:00 am to 5:00 pm.**

*L167: "atmospheric dynamic environment" – justify why dynamic or simplify to atmospheric environment*

**Response: We deleted 'dynamic' in the manuscript.**

*L191: "boundary layer structure regimes " – boundary layer regimes (or structures);*

**Response: We deleted 'structure' in the manuscript.**

*L209: I raised that point before as to me you are calculating Ri in the layer from 0 to z. In your response, you said that you calculate Ri in a thin layer only, although in the text you now say that Ri is calculated at given altitude (not in a layer?). Then you show Eq. 1 and say that theta_vz and theta_v0 are the virtual potential temperatures at the surface and height z, which means that you actually calculate the temperature gradient in the entire layer between 0 and z, contrary to what you said. The same applies to wind: you can only use the values of wind at z knowing that they diminish at 0, otherwise their difference should be used.*
*I strongly suggest that the authors review their approach carefully. The gradient Richardson number formulation includes dz, which in your case is replaced by z, meaning that your layer always has the thickness of z: https://glossary.ametsoc.org/wiki/Gradient_richardson_number*

**Response: We thank the reviewer for pointing out the difference between gradient Richardson number and bulk Richardson number (Zoumakis 1992, Basu et al., 2014). The reviewer is correct that the bulk Richard number is calculated in the entire layer between 0 and Z. From literature, the bulk Richardson number is commonly used for radiosonde-based PLBHT estimations (Vogelezang and Holtslag 1996; Von Engeln and Teixeira, 2013; Chandra et al., 2014; Zhang e t al., 2014). To emphasize that bulk Richardson number is used in this study, we changed $Ri$ to $Ri_b$ and $Ri_c$ to $Ri_{bc}$ in the manuscript.**

**Reference:**

**Basu, S., Holtslag, A.A.M., Caporaso, L. *et al.* Observational Support for the Stability Dependence of the Bulk Richardson Number Across the Stable Boundary Layer. *Boundary-Layer Meteorol* 150, 515–523 (2014). https://doi.org/10.1007/s10546-013-9878-y**

Chandra, S., Dwivedi, A.K. & Kumar, M. Characterization of the atmospheric boundary layer from radiosonde observations along eastern end of monsoon trough of India. *J Earth Syst Sci* 123, 1233–1240 (2014). https://doi.org/10.1007/s12040-014-0458-4

von Engeln, A., & Teixeira, J. (2013). A Planetary Boundary Layer Height Climatology Derived from ECMWF Reanalysis Data, *Journal of Climate*, *26*(17), 6575-6590.

Zoumakis, N.M. On the relationship between the gradient and the bulk Richardson number for the atmospheric surface layer. *Il Nuovo Cimento C* 15, 111–114 (1992). https://doi.org/10.1007/BF02507777

Vogelezang, D.H.P., Holtslag, A.A.M. Evaluation and model impacts of alternative boundary-layer height formulations. *Boundary-Layer Meteorol* 81, 245–269 (1996). https://doi.org/10.1007/BF02430331

Zhang, Y., Gao, Z., Li, D., Li, Y., Zhang, N., Zhao, X., and Chen, J.: On the computation of planetary boundary-layer height using the bulk Richardson number method, Geosci. Model Dev., 7, 2599–2611, https://doi.org/10.5194/gmd-7-2599-2014, 2014.

*L420: If you hypothesize that PBLHT CEIL is greater than PBLHT Liu-Liang because of aerosols, explain what exactly you mean. Are you suggesting that free-tropospheric aerosol may cause some overestimation? If yes then be very specific in your explanation. What do you mean by 'transported aerosol layers'?*

**Response: We rewrite the sentence in lines 410-413 as following:**

**"probably because free-tropospheric aerosol layers transported from low latitudes have larger CEIL backscatter gradients than boundary layer aerosols and the top of the elevated aerosol layer is misidentified as the PBLHT by CEIL."**

*L423: "As being pointed by many previous studies" – this formulation seems grammatically incorrect*

**Response: We thank the reviewer for pointing out the grammar issue. We changed it to 'as was pointed by previous studies' in the manuscript.**

*L427: "The diurnal evolution of PBLHT is important to better understand boundary layer processes." – This sentence is confusing. Diurnal evolution of PBLH is the result of boundary layer processes, but I don't quite see what you want to say here. In the current form, this sentence says that without the diurnal evolution of PBLHT (i.e., for non-evolving PBL) you cannot understand PBL processes, which probably is not the authors intention.*

**Response: We agree with the reviewer. We deleted this sentence because the next sentence is enough to illustrate the point.**

*L483: "Ceilometer provides continuous measurements of aerosol backscatter profiles, which have been widely used to estimate the planetary boundary layer height (PBLHT). Good agreements between ceilometer- and radiosonde-estimated PBLHTs have been reported using*

*limited data from a single location or a short-term campaign. To test the robustness of ceilometer-estimated PBLHT under different atmospheric conditions, we compared ceilometer- and radiosonde-estimated PBLHTs using multiple years of U.S. DOE ARM measurements at six ARM observatories located around the world." – This paragraph needs some polishing (as many others), and I want to show the authors what I mean by that. Please see if my version makes more sense:*

*Ceilometer observations facilitate continuous measurements of aerosol backscatter profiles, which have been widely used to estimate the planetary boundary layer height (PBLHT). Good agreements between the ceilometer and radiosonde estimations have previously been reported for short-term campaigns at single locations. In this study, we extend that comparison to multi-year time series for nine different DOE ARM sites located over land and ocean in different climate zones.*

**Response: We thank the reviewer for showing an example of how to polish the paragraph, which greatly improved the readability of the paragraph. We changed this paragraph using the reviewer's version. We also carefully revised the manuscript with our best effort to improve the readability of the manuscript.**

*L487: "at six ARM observatories located around the world" – This is confusing. Once you say nine (Tab. 1, Fig. 1), then you say six. Again, I strongly encourage the authors to read their paper very carefully and check every single sentence for consistency.*

**Response: We thank the reviewer for pointing out the typo. The reviewer is correct that it should be nine.**

*Figs. 6, 8, 10 – what are the values corresponding to the colors (from red to blue)? Is that scale linear?*

**Response: We thank the reviewer for the comment. The KDE represents the continuous probability density function of observations in datasets and is derived with a bin size of 0.1 km in this study. We now use the same value range of from 0 to 1 in this revision and add a color bar for each figure. The scale is linear.**

*Reply to Reviewer #2's comments*

*Line 187: What is the lowest possible PBLHT from ceilometer? And is an event excluded from the statistics, when an altitude is found below this lowest altitude, also in any of the other methods? Can you include some info on the different vertical resolutions and their impact on the PBLHT (e.g. what happens if all data is smoothed to the same vertical resolution)?*

**Response: Therefore, the CL31 can provide PBLHT estimations from the ground up to 4 km (Münkel et al., 2010). However, since the vertical resolution for PBLHT-SONDE is 30 to 60 m, the minimum PBLHT from PBLHT-SONDE is usually higher than 90 m above the surface. Therefore, we only compare PBLHT higher than 90 m from both PBLHT-SONDE and PBLHT CEIL. We pointed out this in line 363-364.**

**We thank the review for pointing out the impact of different vertical solutions. To reduce the identification of spurious layers due to noisy data, the radiosonde data is subsampled at a 5 mb resolution, corresponding to vertical height resolutions of 30 to 60 meters depending on the atmospheric environment. We added the sentence in the manuscript in lines 166-168. On the other hand, the CL31 applies 30 minutes temporal and 360 m vertical sliding average and provides PBLHT estimations with a vertical resolution of 10 m. Compared with vertical solutions of 30-60 m for PBLHT-SONDE, the CL31 has a higher vertical resolution. We added these discussions in the manuscript in lines 280-281.**

**If all data are smoothed to the same coarser vertical resolution, it is expected the comparisons between PBLTH-SONDE and PBLHT CEIL will be better (in general, the lower the vertical resolution, the better the comparisons). However, this does not mean improved PBLHT estimations from either PBLHT-SONDE or PBLHT CEIL. In addition, it is also technically more complicated to smooth CEIL data to PBLHT-SONDE vertical resolutions as they are dependent on the atmospheric environment.**

*Line 196: Can you include some general info on the percentage within each quality index? How much was removed/used? Any statistical significant removal due to QC? This comment applies to all data sets where QC was used to focus on high quality data.*

**Response: The BL-VIEW algorithm assigns a quality index with a value from 1 to 3 to each boundary layer height candidate. The percentage of each quality index depends on boundary layer aerosol structures. We select the boundary layer height candidate with the highest quality index as the ceilometer estimated PBLHT for each ceilometer profile. Therefore, the quality index is not used to remove PBLHT estimations from the statistics.**

**We do use QC in the PBLHT-SONDE VAP to remove suspicious PBLHT estimations due to bad input data or unreasonable retrievals (e.g., estimated PBLHT > 4 km AGL). Unreasonable PBLHT estimations removed by QC flags are less than 10% of the total data. We added this**

sentence in the manuscript in line 258.

*Line 220: Richardson profile in lower left plot of Figure 3 starts only at about 400m?*

**Response: From Figure 3 c), potential temperatures decrease with height below ~400m, indicating that the atmosphere is superadiabatic near the surface. Therefore, bulk Richardson numbers are negative below ~400m and are not shown in the plot.**

*Line 341: Did you actually check if other bulk Richardson numbers than 0.25 and 0.5 perform better?*

**Response: We thank the reviewer for this suggestion. The magnitude of $Ri_{bc}$ employed in previous studies ranged from 0.25 to 0.5. Seibert et al. (2000) suggests an optimal $Ri_{bc}$ value of 0.25 for the PBLHT estimation when applied to radiosonde data. Therefore, the ARM PBLHT-SONDE Value-Added Product (VAP) provides estimated PBLHTs using two $Ri_{bc}$ values of 0.25 and 0.5. We did not do extra testing of using other bulk Richardson numbers to derive PBLHTs as finding an optimal $Ri_c$ for the PBLHT estimation is out of the scope of this study.**

**Reference:**

**Münkel, C., Roininen, R.: Automatic Monitoring of Boundary Layer Structures with Ceilometer. vol. 184 Vaisala News., 2010.**

**Seibert, P., Beyrich, F., Gryning, S. E., Joffre, S., Rasmussen, A., and Tercier, P.: Review and intercomparison of operational methods for the determination of the mixing height, *Atmos. Environ.,* 34, 1001–1027, https://doi.org/10.1016/S1352-2310(99)00349-0, 2000.**

---

## Author Response (AR3)

*Reply to Reviewer report #1's comments*

*- Line 9: looking at the actual coverage, this is not long term, but multi year. This term is also used appropriately in later parts. Thus, suggest to remove long term.*

**Answer: We changed 'long term' to 'multiple years of' in the text.**

*- Line 254: Just wondering, if LLC and LLC-free add up to 1 in the figure (guess it does, but maybe mention it?)*

**Answer: Yes, LLC and LLC-free occurrence fractions do add up to 1.**

*- Line 284: "SBL boundary layer" change to "SBL"*

**Answer: We changed 'SBL boundary layer' to 'SBL' in the text.**

*- Line 526: Shouldn't this point to figure 2, not 1?*

**Answer: We thank the reviewer for pointing out the typo. We changed 'Figure 1b ' to 'Figure 2b' in the text.**

*- Figure 5b: what is this "Case Amount" actually? I thought I understood it, and then was trying to find out what n is of Figure 6. Judging by the numbers (log scale, so bit difficult to read accurately), the n and the Case Amount are the same thing? So Figure 5b could be remove, as the info in in Figure 6? BUT....*

*- Figure 7: was about to suggest the same as for Figure 5/6 above. But then tried to confirm again, that n and Case Amount are the same thing. However, looking at TWP in 7, it seems slightly above 100. But in Figure 8, it is below 100. So I suggest to define case amount, and if this is n, then remove it from figure 5,7,9 (which seems to agree again with my understanding of n and Case Amount) and only use the n.*

**Answer: We thank the reviewer for the suggestion. We removed the Case Amount plot in figure 5,7, and 9 as suggested. The Case Amount is basically the same as *n* in Figure 6, except that for Figure 5, it is the mean sample numbers for different comparison groups. Each PBLHT-SONDE method has its own QC flag. Samples with no clear QC flag are removed. Therefore, the effective samples for each method are slightly different. That's the reason why the Case Amount in Figure 5 is slightly different with the *n* in Figure 6.**

*Reply to Reviewer report #2's comments*

My general impression is that the new version of the manuscript reads better. However, it still needs some improvements before it can be accepted for publication. Although I previously suggested some changes, it seems that the authors need a stronger and more explicit supervision. I found a couple major problems in this version that I already pointed out in my previous review and the authors didn't seem to address them satisfactorily.

**We are highly appreciated for the reviewer's constructive comments and suggestions, which help to greatly improve the manuscript. We carefully revised the manuscript and tried our best to address the reviewer's comments. The reviewer has a major concern about how Ri is calculated in this study as is presented in the specific comment #12. We want to emphasize that we calculate the 'bulk Richardson number' not Ri. The exact same calculation of 'bulk Richardson number' has been widely used in previous studies for estimating PBLHT. We also modified the definition sentence of $Ri_b$ to emphasize that it is calculated 'between the surface and a given altitude'.**

Specific comments:

1. In my second review, I recommended that the authors read the paper carefully again and check every single sentence for consistency. The authors' response was:
   "We carefully revised the manuscript according to the reviewer's comments."
   When I started reading the new version of the manuscript, the first thing that struck me was the abstract, in which you say:"... at three ARM fixed-location atmospheric observatories and from three ARM mobile observatories deployed around the world for various field campaigns, which cover from Tropics to Polar regions and over both ocean and land surfaces." and that information is then repeated in the Introduction.

   According to the above, you have six sources of data in total. Just to remind you one of the points from my previous review:
   *L487: "at six ARM observatories located around the world" – This is confusing. Once you say nine (Tab. 1, Fig. 1), then you say six. Again, I strongly encourage the authors to read their paper very carefully and check every single sentence for consistency.*

   *Response: We thank the reviewer for pointing out the typo. The reviewer is correct that it should be nine.*

   Clearly, it was an error not a typo and that error was then repeated in the abstract and the Introduction and is still there. It is difficult for me to understand what the authors mean by "we carefully revised the manuscript" since I find the same simple errors again.

   **Answer: we apologize for causing the confusion. In line 118, we pointed that 'ARM operates three mobile facilities (AMFs) which can be requested by scientists through a proposal process for various field campaigns that deploy ARM instruments anywhere in the world for roughly a year'. Each ARM mobile facility could be deployed at a**

**location for a field campaign and then moved and deployed at another location for a difference field campaign. The three ARM mobile facilities were deployed at various locations around the world for field campaigns during the past decade.**

**To avoid the confusion, we delete the two 'three's in the abstract.**

"which cover from Tropics to Polar regions" – cover what? Use lowercase.

**Answer: we changed 'which' to 'These observatories' and used lowercase for 'the tropics' and 'the polar regions' in the text.**

I suggest to remove the number of stations from the abstract and describe them in a more general way, for instance as a data set from different climate zones probing a variety of PBL regimes.

**Answer: we removed the number of stations from the abstract as suggested. Detailed descriptions of ARM observations are presented in Section 2.**

2. Title: should it be "with" or "and"? The current form suggests that radiosonde data is good and ceilometer data needs to be evaluated.

   **Answer: we agree with the reviewer that PBLHT from radiosonde data is good, so we use 'with' in the title.**

3. L30: Not true. PBL height does not characterize the structure of the lowest few kilometers of the atmosphere by any means. It only indicates where the top of that PBL structure is located.

   **Answer: we deleted this sentence to avoid the confusion.**

4. L36-37: "by an inversion layer of potential temperature" – this sentence is unclear. Is it a temperature inversion? Why "inversion layer"? For convective PBLs potential temperature actually does not have an inversion at the top as its stratification changes from neutral to stable, right? Did you mean real temperature, for which temperature inversion can make more sense?

   **Answer: we thank the reviewer for pointing it out. It should be 'temperature'. We deleted 'potential' in the sentence.**

5. There are at least 9 different methods of estimating PBL height. Von Engeln and Teixeira (2013) mention many of them. In large-eddy simulations, it is common to apply gradient methods (for temperature or moisture) or indeed look at turbulence properties. Some examples worth citing:

   Bopape, M.-J.M.; Plant, R.S.; Coceal, O. Resolution Dependence of Turbulent Structures in Convective Boundary Layer Simulations. *Atmosphere* **2020**, *11*, 986. https://doi.org/10.3390/atmos11090986

J. Kurowski, M., P. Malinowski, S. and W. Grabowski, W. (2009), A numerical investigation of entrainment and transport within a stratocumulus-topped boundary layer. Q.J.R. Meteorol. Soc., 135: 77-92. https://doi.org/10.1002/qj.354

**Answer: we thank the reviewer for pointing out these previous and recent works. We added references to them.**

6. L84: ", the robustness…" – this should either be a new sentence or you should add "and" in between

**Answer: we started a new sentence as suggested.**

7. L87: "long term" – this is too vague and means different things for different people; be more specific: multi-year?

**Answer: we replaced 'long term' with 'multiple years of' in the text.**

8. L97: "low-level cloud-free" – what are low level cloud free conditions here? What does that low level refer to? Does it mean there are clouds at some upper levels?

**Answer: we defined the 'low-level cloud-free' condition in the line 273 at the 'Results and Discussions' section as: "comparisons are also separated for conditions with and without the presence of low-level clouds below 4 km AGL (referred to as LLC and LLC-free, correspondently), as detected by the ceilometer at the time of the radiosonde launch."**

9. L131-132: "three commonly used methods developed by Heffter (1980; referred to as the Heffter method hereafter)" – does it mean you name the three methods as the Hefter method?

**Answer: we rewrote the sentence as:**

**'The ARM PBLHT-SONDE VAP implements three commonly used methods including the Heffter method (Heffter 1980), the Liu and Liang method (Liu and Liang 2010), and the bulk Richardson number method (Seibert et al., 2000) to estimate PBLHT from radiosonde data (Sivaraman et al., 2013).'**

10. That description of 3 methods is still slightly chaotic.
Please summarize the text between L130 and L155 with an additional table: Method name, algorithm (e.g., Richardson number, potential temperature gradient), reference, etc. which should help understand your methodology.

**Answer: as suggested, we added a table to summarize the algorithms that are used to estimate PBLHT.**

11. L145: when you say between fifth and second levels of sounding data it means nothing to the reader. Explain what levels 5 and 2 mean. Is it at a fixed height which is the same at all stations

under all conditions?
In principle, if you use a threshold for potential temperature difference, it means you assume that there must exist a minimum gradient. So that method also relies on temperature gradient.

**Answer: Following Liu and Liang (2010), the potential temperature ($\theta$) difference between the fifth and second level of sounding data ($\theta_5$ - $\theta_2$) is used to represent the near-surface thermal gradient. We added this illustration in the text. ($\theta_5$ - $\theta_2$) might correspond to slightly different heights and pressures for different locations due to different site altitudes and local atmospheric environments. In line 179, we pointed out that 'the radiosonde data is subsampled at a 5 mb resolution, corresponding to vertical height resolutions of 30 to 60 m depending on the atmospheric environment'. It is true that the threshold might be affected by multiple factors. We pointed out in line 198 that 'threshold values of $\delta_s$, $\delta_u$, and $\dot{\theta}_r$ are dependent on the surface type and are empirically determined in Liu and Liang (2010).' The method is not perfect for determining PBL regimes, but it provides a practical way to estimate PBL regimes using radiosonde data.**

12. L160: We need to discuss this definition of Ri one more time. According to your description: "Ri at a given altitude can be calculated from sounding data"

$$Ri_b = \left(\frac{gz}{\theta_{v0}}\right)\left(\frac{\theta_{vz}-\theta_{v0}}{u_z^2+v_z^2}\right)$$

As I explained in my previous reviews, this particular definition refers to the entire layer from 0 (surface) up to some height z. This is not Ri at a given altitude and you can only do it for a layer of finite height. If you want to look at a layer of thickness dz, then it should rather be:

$$R_B = \frac{(g/T_v)\Delta\theta_v\Delta z}{(\Delta U)^2 + (\Delta V)^2}$$

Because your definition replaces all the gradients with the values at the height z, it implicitly assumes that those values diminish at z=0 (which is true for u and v, but not for theta that uses a difference between level 0 and z in your equation. How exactly is Ri calculated in your data?

If you still claim that Ri in your equation (1) is given at the height z, then we have a problem with understanding basic equations used in your study. I tried to explain it in my previous reviews, but was so far unsuccessful.
It is critically important that you understand and explain the way you use Ri in your study correctly as it is the foundation of your analysis.

**Answer: We thank the reviewer for the suggestion. We used the exact same definition of 'bulk Richardson number (Ri$_b$)' as was widely used in previous studies such as the Equation (7) in Sørensen et al., (1998), the Equation (9) in Seibert et al., (2000), the**

Equation (1) in Zhang et al., (2014), and the Equation (1) in Bakas et al., (2020), etc. We calculated and used 'bulk Richardson number ($Ri_b$)' not Ri. To avoid such confusion, we changed the definition to '$Ri_b$ between the surface and a given altitude $z$ can be calculated from …' in the line 213 in the text. The equation is also consistent with the $R_B$ equation the author suggested given that the PBL is the lowest layer of the atmosphere. Although a critical value ($Ri_{bc}$) is not well defined for the bulk Richardson number ($Ri_b$), Seibert et al. (2000) suggests an optimal $Ri_{bc}$ value of 0.25 when applied to radiosonde data. All variables on the right side in the equation (1) can be obtained from radiosonde data and therefore it is straight forward to calculated 'the bulk Richardson number' from radiosonde data.

**References:**

Bakas, N.A.; Fotiadi, A.; Kariofillidi, S. Climatology of the Boundary Layer Height and of the Wind Field over Greece. *Atmosphere* 2020, *11*, 910. https://doi.org/10.3390/atmos11090910

Seibert, P., Beyrich, F., Gryning, S. E., Joffre, S., Rasmussen, A., and Tercier, P.: Review and intercomparison of operational methods for the determination of the mixing height, *Atmos. Environ.,* 34, 1001–1027, https://doi.org/10.1016/S1352-2310(99)00349-0, 2000.

Sørensen, J.H., Rasmussen, A., Ellermann, T. and Lyck, E.: Mesoscale Influence on Long-range Transport – Evidence From ETEX Modeling and Observations, *Atmospheric Environment,* 32(24): 4207–4217, https://doi.org/10.1007/978-1-4615-4153-0_27, 1998.

Zhang, Y., Gao, Z., Li, D., Li, Y., Zhang, N., Zhao, X., and Chen, J.: On the computation of planetary boundary-layer height using the bulk Richardson number method, Geosci. Model Dev., 7, 2599–2611, https://doi.org/10.5194/gmd-7-2599-2014, 2014.

13. L265: "because Liu-Liang method uses different methods" – confusing

    **Answer: We changed it to 'because the Liu-Liang method uses different algorithms …as discussed in section 2' to make it clearer.**

14. Fig. 11: Make axes labels and ticks larger.

    **Answer: We thank the reviewer for the suggestion. We use larger axes labels and ticks in the Figure.**